# Magic spreading in random quantum circuits

Xhek Turkeshi [1] ✉, Emanuele Tirrito[2,3] & Piotr Sierant [4,5] ✉

Magic is the resource that quantifies the amount of beyond-Clifford operations necessary for universal quantum computing. It bounds the cost of classically simulating quantum systems via stabilizer circuits central to quantum error correction and computation. In this paper, we investigate how fast generic many-body dynamics generate magic resources under the constraints of locality and unitarity, focusing on magic spreading in brick-wall random unitary circuits. We explore scalable magic measures intimately connected to the algebraic structure of the Clifford group. These metrics enable the investigation of the spreading of magic for system sizes of up to $N = 1024$ qudits, surpassing the previous state-of-the-art, which was restricted to about a dozen qudits. We demonstrate that magic resources equilibrate on timescales logarithmic in the system size, akin to anti-concentration and Hilbert space delocalization phenomena, but qualitatively different from the spreading of entanglement entropy. As random circuits are minimal models for chaotic dynamics, we conjecture that our findings describe the phenomenology of magic resources growth in a broad class of chaotic many-body systems.

Quantum computers require several types of resources to solve computational tasks faster than classical computers[1,2]. Entanglement is one such resource, but alone, it is insufficient to guarantee that a quantum computer outperforms its classical counterpart. Indeed, stabilizer states can attain extensive entanglement under Clifford operations while being efficiently simulatable on classical computers via the Gottesman-Knill theorem[3–5]. Nonstabilizerness, colloquially called "magic", quantifies the additional non-Clifford operations required to perform a given quantum operation, constituting another necessary ingredient for the quantum speedup[6]. Understanding how magic resources build up and propagate in many-body quantum systems emerges as a fundamental question, with potential impact on current and near-term quantum devices[7].

The question of magic resources generation is ambitious but challenging. Until a few years ago, measures of magic required minimization procedures over large spaces, resulting in prohibitive computational costs for even a few qubits[8]. Recently, mana and stabilizer entropies have been introduced as scalable measures of magic[9–13]. Subsequent developments in tensor network methods[14–17] and Monte-Carlo approaches[18,19] have provided a powerful toolbox to characterize the nonstabilizerness of ground states[20] while enabling hybrid Clifford-tensor network

algorithms[21–25]. Despite these successes, the time evolution of magic resources in many-body systems remains a largely open question. The rapidly growing entanglement limits the traditional tensor network methods and brute-force exact simulations to small system sizes. With these limitations, Ref. 26 concluded that a quantum quench in an integrable system results in a linear growth of nonstabilizerness over time, similar to what happens for the entanglement entropy[27].

This work investigates the magic spreading under generic, nonintegrable, local unitary quantum dynamics. To that end, we focus on Haar random brick-wall circuits of qudits. Unambiguous identification of the features of the growth of magic resources requires access to large system sizes. For this reason, inspired by the algebraic structure of the Clifford group, we consider the family of generalized stabilizer entropies (GSE). The latter constitute good measures of nonstabilizerness for many-body systems and include stabilizer Rényi entropy (SRE) as a particular example. We combine the replica trick and Haar average methods to express the circuit-averaged GSE as tensor network contractions. In particular, the GSE can be expressed as a $k = 3$ replica quantity for qutrits while necessitating $k = 4$ replica for qubits. In both cases, the replica tensor network methods allow us to investigate

[1]Institut für Theoretische Physik, Universität zu Köln, Köln, Germany. [2]The Abdus Salam International Centre for Theoretical Physics (ICTP), Trieste, Italy. [3]Pitaevskii BEC Center, CNR-INO and Dipartimento di Fisica, Università di Trento, Trento, Italy. [4]ICFO-Institut de Ciéncies Fotòniques, The Barcelona Institute of Science and Technology, Castelldefels, Barcelona, Spain. [5]Barcelona Supercomputing Center, Barcelona, Spain. ✉e-mail: turkeshi@thp.uni-koeln.de; piotr.sierant@bsc.es

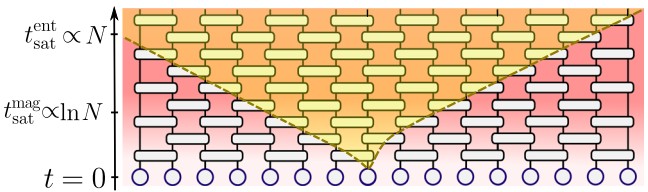

**Fig. 1 | Cartoon of magic spreading in random circuits.** A system of $N$ qudits is prepared at time $t = 0$ in a product state $|\Psi_0\rangle$ with low-magic resources. The evolution under a quantum circuit comprising local Haar-random gates increases the nonstabilizerness of the state (denoted by the red gradient) and scrambles quantum information (symbolized by the light-cone). The nonstabilizerness approaches its long-time saturation value up to a given tolerance $\epsilon \ll 1$ at time $t_{\text{sat}}^{\text{mag}} \propto \ln N$, scaling logarithmically with $N$, while distant qudits become entangled only after a longer time $t_{\text{sat}}^{\text{ent}} \propto N$.

systems of up to $N \leq 1024$ qubits. Our main finding is that the long-time saturation value of GSE entropy is reached, up to a tolerance $\epsilon \ll 1$, at times $t_{\text{sat}}^{\text{mag}} \propto \ln N$, scaling logarithmically with system size, see Fig. 1.

## Results

### Generalized stabilizer entropies

We start by discussing the GSE, characterizing the magic resources for systems of $N$ qudits[10,28]. Denoting $\mathbb{Z}_d = \{0, 1, \ldots, d - 1\}$ the finite field with $d$ elements, the local Hilbert space of a $d$-dimensional qudit is $\mathcal{H}_d = \text{span}[\{|m\rangle\}_{m \in \mathbb{Z}_d}]$. The GSEs are tied to the algebraic structure of the Pauli and Clifford groups[3] acting on $\mathcal{H}_d^{\otimes N}$. Let us denote the qudit Pauli operators

$$X = \sum_{m=0}^{d-1} |m\rangle\langle m \oplus_d 1|, \qquad Z = \sum_{m=0}^{d-1} \omega^m |m\rangle\langle m|, \qquad (1)$$

with $a \oplus_d b = a + b \mod d$ representing the sum in $\mathbb{Z}_d$ and $\omega = e^{2\pi i/d}$[29]. Then, the set of Pauli strings $\mathcal{P}_N(d) = \{X_1^{r_1^x} Z_1^{r_1^z} X_2^{r_2^x} Z_2^{r_2^z} \cdots X_N^{r_N^x} Z_N^{r_N^z} \mid r_k^\alpha \in \mathbb{Z}_d\}$ is generated by tensor products of Pauli operators. The Clifford group $\mathcal{C}_{N,d}$ consists of the unitary $C$ mapping, up to a global phase, a Pauli string $P$ to a Pauli string $\omega^r P' = CPC^\dagger$ with $r \in \mathbb{Z}_d$. Stabilizer states are defined as $\text{STAB}_{N,d} = \{C|0\rangle^{\otimes N} \mid C \in \mathcal{C}_{N,d}\}$, and magic or non-stabilizer states are those not belonging to this set.

The theory of GSE is intimately connected to the structure of the commutants of the Clifford group for $k$ copies of the system[30,31], cf. Supplementary Note 1 for a brief review. For a given set $\mathcal{E}$, we define its $k$-th commutant as the set of operators $W$ acting on $k$ copies of the system such that $\text{Comm}_k(\mathcal{E}) = \{W \mid [W, E^{\otimes k}] = 0 \text{ for all } E \in \mathcal{E}\}$. For the unitary group $\mathcal{U}(d^N)$, the commutant is built of the representation of permutation operators $\text{Comm}_k(\mathcal{U}(d^N)) = \{W_\pi \mid \pi \in S_k\}$[32]. By duality, since $\mathcal{C}_{N,d} \subset \mathcal{U}(d^N)$, it follows that $\text{Comm}_k(\mathcal{U}(d^N)) \subset \text{Comm}_k(\mathcal{C}_{N,d})$. Importantly, the Clifford commutant contains $|\text{Comm}_k(\mathcal{C}_{N,d})| = \prod_{m=0}^{k-2}(d^m + 1)$ elements, whereas $|\text{Comm}_k(\mathcal{U}(d^N))| = k!$. These two numbers coincide for $d > 2$ when $k \leq 2$, and for $d = 2$ when $k \leq 3$. Hence, the elements of the commutant, when applied to a state $(|\Psi\rangle\langle\Psi|)^{\otimes k}$ cannot distinguish whether $|\Psi\rangle$ is a stabilizer state or whether it has non-vanishing magic resources if $k \leq 3$ for qubits and $k \leq 2$ for $d \geq 3$.

On the other hand, when $k \geq 3$ for qudits (and $k \geq 4$ for qubits), the Clifford commutant is strictly greater than $\text{Comm}_k(\mathcal{U}(d^N))$. In that case, the intrinsic Clifford commutant[28], defined as $\overline{\text{Comm}}_k(\mathcal{C}_{N,d}) = \text{Comm}_k(\mathcal{C}_{N,d}) \setminus \text{Comm}_k(\mathcal{U}(d^N))$, is a non-empty set, whose elements $W$, when applied to $(|\Psi\rangle\langle\Psi|)^{\otimes k}$ can distinguish if the state is a stabilizer state or not. We define the generalized stabilizer entropy, $M_W$, and the associated generalized stabilizer purity $\zeta_W$ by

$$M_W \equiv -\ln[\zeta_W(|\Psi\rangle)], \quad \zeta_W \equiv \text{tr}(W|\Psi\rangle\langle\Psi|^{\otimes k}). \qquad (2)$$

For any $W \in \overline{\text{Comm}}_k(\mathcal{C}_{N,d})$, the generalized stabilizer purity $\zeta_W(|\Psi\rangle) \leq 1$, with the equality holding if and only if $|\Psi\rangle$ is a stabilizer state, as we show

in the Supplementary Note 2. This implies that the GSE $M_W(|\Psi\rangle) \geq 0$, with the equality holding if and only if $|\Psi\rangle$ is a stabilizer state. Moreover, for any operator $W$ from the intrinsic Clifford commutant, the associated GSE $M_W$ is a measure of magic for many-body systems: (i) $M_W(|\Psi\rangle) \geq 0$ and $M_W = 0$ if and only if $|\Psi\rangle$ is a stabilizer state, (ii) $M_W(C|\Psi\rangle) = M_W(|\Psi\rangle)$ for $C \in \mathcal{C}_{N,d}$ a Clifford unitary, (iii) it is additive $M_W(|\Phi\rangle \otimes |\Psi\rangle) = M_W(|\Phi\rangle) + M_W(|\Psi\rangle)$. The subclass of stabilizer Rényi entropies are monotones for generic stabilizer protocols for qubits[12]. A key contribution of our work is showing that stabilizer entropies are monotone also for prime dimension qudits ($d \geq 3$), as we discussed in Methods. In the Supplementary Note 3 we also argue that monotonicity under Pauli measurements of arbitrary GSE is not guaranteed, presenting an explicit counterexample for qutrits systems.

To understand better the scope of GSE, we first consider how SRE arise from (2). At $k = 4$, the intrinsic Clifford commutant contains several operators, one of which is $Q_4 = \sum_{P \in \mathcal{P}_N(d)}(P \otimes P^\dagger)^{\otimes 2}/d^N$. This operator leads to the second SRE[10] $M_2 \equiv M_{Q_4} = -\ln[\zeta_{Q_4}]$ with $\zeta_{Q_4} = \sum_{P \in \mathcal{P}_N(d)} |\langle\Psi|P|\Psi\rangle|^4/d^N$. Similarly, the SRE $M_\alpha$ of arbitrary integer index $\alpha \geq 2$, $M_\alpha \equiv \ln[\sum_P |\langle\Psi|P|\Psi\rangle|^{2\alpha}/d^N]/(1 - \alpha)$, fulfills $M_\alpha \propto M_{Q_{2\alpha}}$ with $Q_{2\alpha} = \sum_{P \in \mathcal{P}_N(d)}(P \otimes P^\dagger)^{\otimes \alpha}/d^N$. With a slight abuse of notation, throughout this text, we will consider $M_2 \equiv M_{Q_4}$. An example of GSE beyond the family of SREs is by the operator $Y_d \equiv \sum_{P \in \mathcal{P}_N(d)}(P \otimes P \otimes P^{d-2})/d^N$, which belongs to the intrinsic Clifford commutant for $k = 3$ replicas in any qudit system with $d \geq 3$ prime. Throughout this paper, we denote by $M_Y$ the GSE induced by the operator $Y_d$, with the dimension $d$ inferred from context.

The above examples provide concrete measures of the magic $M_W$ that require $k = 3$ copies for qudits with odd prime $d$ and $k = 4$ for qubits. A figure of merit valid for any stabilizer purity $\zeta_W$ is their efficient tensor network representation. In fact, the Clifford commutant operators $T$ acting on $N$ qudit systems, reduce to tensor products of operators $W = w^{\otimes N}$ acting on individual qudits. As a result, Eq. (2) is efficiently computable via tensor network methods, either by exact contractions[14,17] or by sampling methods[16], see Supplementary Note 4 for details.

### Brick-wall Haar random quantum circuits

We consider a one-dimensional chain of $N$ qudits and study the spreading of the GSEs $M_W$ under unitary dynamics generated by brick-wall Haar random quantum circuits (see Fig. 1). The evolution operator of the considered brick-wall circuit reads $U_t = \prod_{r=1}^t U^{(r)}$, where $t$ is the circuit depth – also referred to as time. Numbering the qudits by $i = 1, \ldots, N$, the layers $U^{(r)}$ are fixed as

$$U^{(2m)} = \prod_{i=1}^{N/2-1} U_{2i, 2i+1}, \quad U^{(2m+1)} = \prod_{i=1}^{N/2} U_{2i-1, 2i}, \qquad (3)$$

comprising two-qubit gates $U_{ij}$ chosen independently with the Haar distribution on the unitary group $\mathcal{U}(d^2)$. The initial state $|\Psi_0\rangle$ is chosen as the stabilizer state $|\Psi_0\rangle = |0\rangle^{\otimes N}$, with $M_W = 0$ for any $W$ in the intrinsic Clifford commutant. How do the magic resources of $|\Psi_t\rangle$, quantified by the GSEs, increase under the dynamics of the circuit (3)?

The problem at hand is stochastic due to the randomness of the gates. Denoting with $\mathbb{E}(\bullet)$ the average over the circuit realizations, we consider quenched and annealed averages of the GSEs, defined respectively as $\overline{M_W} \equiv \mathbb{E}[M_W(|\Psi_t\rangle)] = -\mathbb{E}[\ln[\zeta_W(|\Psi_t\rangle)]]$ and $\tilde{M}_W \equiv -\ln[\mathbb{E}[\zeta_W(|\Psi_t\rangle)]]$. Exact numerical simulation of the random circuit (3) provides access to $|\Psi_t\rangle$, allowing for calculation of the quenched and the annealed averages of the GSEs and exposing the self-averaging of $\zeta_W(|\Psi_t\rangle)$. The circuit-to-circuit fluctuations of $\zeta_W(|\Psi_t\rangle)$ around its average value $\mathbb{E}[\zeta_W(|\Psi_t\rangle)]$ are suppressed with the increase of the system size $N$ and decay rapidly in time as discussed in Methods for several choices of $W$. The self-averaging of $\zeta_W(|\Psi_t\rangle)$ implies that $\overline{M_W}$ and $\tilde{M}_W$ approach each other with increase of $N$ and $t$. Therefore, the annealed average $\tilde{M}_W$ may be chosen to quantify the time evolution of the GSEs under the random circuits.

## Annealed average of generalized stabilizer entropies

The calculation of the annealed average $\bar{M}_W$ is facilitated by a replica trick and the Weingarten calculus, which allow us to map computation of $\bar{M}_W$ for the random Haar circuits to a contraction of a two-dimensional tensor network. The latter can be efficiently computed to provide insights into the time evolution of the GSEs for systems comprising hundreds of qudits, far beyond the reach of exact simulation of the system.

For convenience, we employ the superoperator formalism[33]: $A \mapsto |A\rangle\rangle$, $U_t A U_t^\dagger \mapsto (U_t \otimes U_t^*)|A\rangle\rangle$, and $\langle\langle A|B\rangle\rangle = \mathrm{tr}(A^\dagger B)$. Using the common abuse of notation of implicit reshaping $(\mathcal{H}_d^{\otimes N})^{\otimes k} \mapsto (\mathcal{H}_d^{\otimes k})^{\otimes N}$ when necessary, we have

$$\zeta_W(|\Psi_t\rangle) = \langle\langle W|(U_t \otimes U_t^*)^{\otimes k}|\rho_0^{\otimes k}\rangle\rangle, \qquad (4)$$

where $\rho_0 = |\Psi_0\rangle\langle\Psi_0|$ is the initial state's density matrix. To calculate the average $\mathbb{E}[\zeta_W(|\Psi_t\rangle)]$ over the circuit realizations, we observe that (4) is linear in $(U_t \otimes U_t^*)^{\otimes k}$, implying that we can first average the superoperator corresponding to the circuit and then calculate the matrix element in (4). Due to the statistical independence of the two-body gates at various spatial and temporal locations, the former reduces to evaluating the two qudit transfer matrix $\mathcal{T}_{i,i+1}^{(k)} \equiv \mathbb{E}_{\mathrm{Haar}}[(U_{i,i+1} \otimes U_{i,i+1}^*)^{\otimes k}]$. These expressions are formulated in terms of the unitary commutant $\mathrm{Comm}_k(\mathcal{U}(d^N))$ which, as aforementinoed, consists of permutation operators. Up to reshaping, we express $\mathcal{T}^{(k)}$ via the corresponding permutation states $|\tau\rangle\rangle_i$ acting on the $k$-replica qudits at sites $i, i+1$, leading to the expression

$$\mathcal{T}_{i,i+1}^{(k)} = \sum_{\pi,\tau \in S_k} \mathrm{Wg}_{\tau,\sigma}(d^2)|\tau\rangle\rangle_i|\tau\rangle\rangle_{i+1}\langle\langle\sigma|_i\langle\langle\sigma|_{i+1}, \qquad (5)$$

where $\langle\langle b_1, \bar{b}_1, \ldots, b_k, \bar{b}_k|\tau\rangle\rangle = \prod_{m=1}^k \delta_{b_m, \bar{b}_{\tau(m)}}$ for each $k$-replica qudit basis state $|b_1, \bar{b}_1, \ldots, b_k, \bar{b}_k\rangle\rangle$ and $\mathrm{Wg}_{\tau,\sigma}(d^2)$ denotes the Weingarten symbol[34]. The lattice structure induced by the circuit requires contraction of $\mathcal{T}_{i,i+1}^{(k)}$ between the even and odd layers (3), with the overlaps $G_{\sigma,\tau}(d) \equiv \langle\langle\sigma|\tau\rangle\rangle = d^{\#(\sigma^{-1}\tau)}$ taken into account, where $\#(\tau)$ denotes the number of cycles for $\tau \in S_k$. We reabsorb these overlaps by defining the tensors

$$\mathcal{T}_{i,i+1}^{(k)} \equiv \blacksquare \equiv \sum_{\pi_1,\pi_2,\pi,\tau \in S_k} \mathrm{Wg}_{\tau,\pi}(d^2) \times$$
$$G_{\pi,\pi_1}(d)G_{\pi,\pi_2}(d)|\tau\rangle\rangle_i|\tau\rangle\rangle_{i+1}\langle\langle\hat{\pi}_1|_i\langle\langle\hat{\pi}_2|_{i+1}, \qquad (6)$$

with the states $|\hat{\sigma}\rangle\rangle$ satisfying $\langle\langle\hat{\sigma}|\tau\rangle\rangle = \delta_{\sigma,\tau}$. The contraction with the first layer of unitary gates is fixed by the replica boundary condition

$$\boxed{+} \equiv \mathcal{T}_{i,i+1}^{(k)}|\rho_0^{\otimes k}\rangle\rangle = \sum_{\pi \in S_k} \frac{(d^2-1)!}{(d^2+k-1)!}|\pi\rangle\rangle_i|\pi\rangle\rangle_{i+1}, \qquad (7)$$

while the contraction with the last layer of the circuit requires

$$\boxed{\mathfrak{w}} \equiv \langle\langle w|_i\langle\langle w|_{i+1}\mathcal{T}_{i,i+1}^{(k)}. \qquad (8)$$

Summarizing, the computation of the annealed average of the GSEs reduces to evaluating the tensor contraction

$$\mathbb{E}[\zeta_W(|\Psi_t\rangle)] = \qquad (9)$$

The effective "spins", i.e., the degrees of freedom at the sites of the lattice (9), correspond to permutations of the $k$ replicas and hence admit $q_{\mathrm{eff}} = k!$ values, while the tensors $\mathcal{T}_{i,i+1}^{(k)}$ can be interpreted as non-unitary gates acting on the spins. These observations constitute the basis of our numerical approach, which allows us to compute the annealed average of the GSEs $\bar{M}_W = -\ln[\mathbb{E}(\zeta_W(|\Psi_t\rangle))]$ for arbitrary circuit depth $t$. While the above discussion applies to any intrinsic Clifford commutant operator $W$, we will specifically analyze the SRE $M_2$ for $k = 4$ replicas in qubit ($d = 2$) and qutrit ($d = 3$) systems, and $M_Y$ for $k = 3$ in qutrit systems.

## Deep circuit limit

In the deep circuit limit, for $t \gg 1$, the brick-wall quantum circuits form approximate $k$-designs[35,36]. In that limit, the operator $U_t$ in (4) can be replaced by a global Haar random gate $U \in \mathcal{U}(d^N)$ and $\mathcal{T}_{i,i+1}^{(k)}$ in the contraction (9) are substituted by the global gate. This allows for analytical calculation of the $\bar{M}_W$ of interest as detailed in the Supplementary Note 5, yielding $M_2^{\mathrm{Haar}} \equiv -\ln[4/(2^N + 3)]$ for $d = 2$[37], while for $d = 3$, we find $M_2^{\mathrm{Haar}} = M_Y^{\mathrm{Haar}} \equiv -\ln[3/(3^N + 2)]$. Due to the concentration of Haar measure[32], the fluctuations of $M_W$ with circuit realizations are strongly suppressed with the increase of $N$. Hence, the GSEs $\bar{M}_W$ saturate at long times $t \gg 1$ under the dynamics of random circuits to $M_W^{\mathrm{Haar}}$. Now, we characterize the approach of $M_W(|\Psi_t\rangle)$ to the saturation value $M_W^{\mathrm{Haar}}$.

## Numerical results

Our results for the growth of GSEs under the dynamics of random circuits are summarized in Figs. 2–4 for qubits ($d = 2$) and qutrits ($d = 3$), respectively. We start by comparing the quenched $\tilde{M}_W$ and annealed $\bar{M}_W$ averages of the GSEs. As anticipated, already for $N = 8$ qubits and qudits, we find $\tilde{M}_W \approx \overline{M}_W$, confirming that the quenched and annealed averages can be used interchangeably to characterize magic spreading in the considered circuits, cf. Methods for details. Hence, we focus on the annealed averages $\bar{M}_W$ obtained from the tensor network contraction (9). Expressing the state of the $q_{\mathrm{eff}} = k!$ dimensional "spins" as a matrix product state[38], we contract the tensor network (9) horizontally, layer after layer. Implementing the contraction in ITensor[39], we observe that a bond dimension $\chi = \mathcal{O}(q_{\mathrm{eff}}^2)$ of the matrix product state is sufficient to obtain converged results, see Methods. The computation requires significantly smaller resources for qutrits since $M_Y$ is a $k = 3$-replica quantity resulting in $q_{\mathrm{eff}} = 6$. In contrast, the $k = 4$ replicas demanded to calculate $M_2$ for qubits lead to $q_{\mathrm{eff}} = 24$, and significantly larger computational costs. In this case, to alleviate computational complexity, we use irreducible representations of the permutation group, effectively reducing the replica degrees of freedom to $q_{\mathrm{eff}} = 14$[40]. We compute $\tilde{M}_2$ for systems of up to $N = 1024$ qubits ($N = 512$ qutrits) with $\chi = 300$ ($\chi = 800$), as shown in Fig. 2 (Fig. 3), and $\tilde{M}_Y$ for $N \leq 1024$ qutrit systems with $\chi = 300$, cf. Fig. 4.

In all the cases, the GSEs $\bar{M}_W$ are proportional to the system size $N$ already at $t = 1$. Indeed, the additivity of $\bar{M}_W$ implies that $\bar{M}_W(|\Psi_{t=1}\rangle) = (N/2)\tilde{M}_W^{(2)}$, where $\tilde{M}_W^{(2)}$ is the average GSE generated by a single two-body gate $U_{i,i+1}$, and $N/2$ is the number of two-body gates in the first layer of the circuit. For $t > 1$, the GSEs $\bar{M}_W$ rapidly increase towards their saturation values $M_W^{\mathrm{Haar}}$ for both $d = 2$ and $d = 3$. For circuit depths $t \gtrsim 5$, the difference $\Delta M_W(t) = M_W^{\mathrm{Haar}} - \bar{M}_W(|\Psi_t\rangle)$ is proportional to the system size $N$ and decays exponentially in time:

$$\Delta M_d(t) = a_{d,W} N e^{-\alpha_{d,W} t}, \qquad (10)$$

where $a_{d,W}$ and $\alpha_{d,W}$ are constants (see Figs. 2–4). The exponential relaxation of the GSEs to their long-time saturation values under the dynamics of random quantum circuits is the main result of this work. The saturation value of GSEs is reached, up to a fixed small accuracy $\epsilon$, i.e., $\Delta M_W = \epsilon$, at time $t_{\mathrm{sat}}^{\mathrm{mag}} = \ln(N)/\alpha_{d,W} + O(1)$, scaling logarithmically with system size $N$.

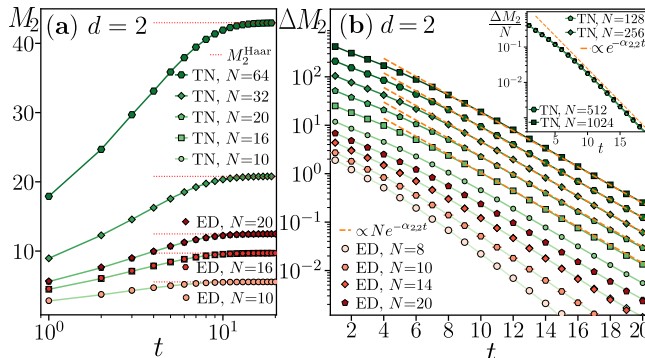

**Fig. 2 | Stabilizer Rényi entropy evolution in qubit circuits. a** The SRE $M_2$ abruptly saturates to $M_2^{\text{Haar}}$. **b** The difference $\Delta M_2 = M_2^{\text{Haar}} - M_2$ approaches exponential decay $\Delta M_2 \propto N e^{-\alpha_{2,2}t}$, where $\alpha_{2,2} = 0.43(3)$, see the inset. The annealed average $\bar{M}_2$ obtained via (9) (denoted "TN") and the quenched average $\overline{M}_2$ (denoted "ED") coincide within the error bars already for $N = 8$.

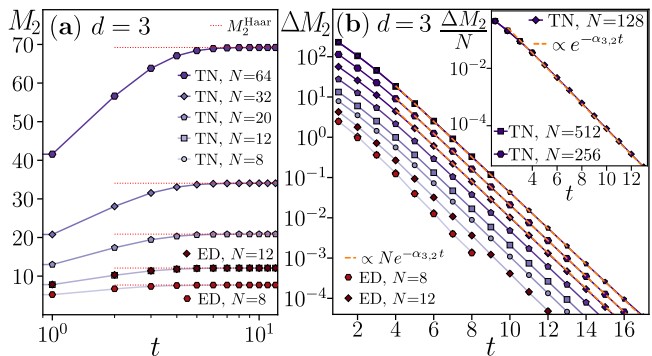

**Fig. 3 | Stabilizer Rényi entropy evolution in qutrit circuits. a** The saturation of $M_2$ to $M_2^{\text{Haar}}$ for $d = 3$ occurs similarly to the qubit case. **b** The difference $\Delta M_2 = M_2^{\text{Haar}} - M_2$ follows $\Delta M_2 \propto N e^{-\alpha_{3,2}t}$ with $\alpha_{3,2} = 1.03(3)$ at $t \gtrsim 5$; see the inset. The quenched $\bar{M}_2$ and annealed $\overline{M}_2$ averages are indistinguishable on the scale of the figure for any $N$.

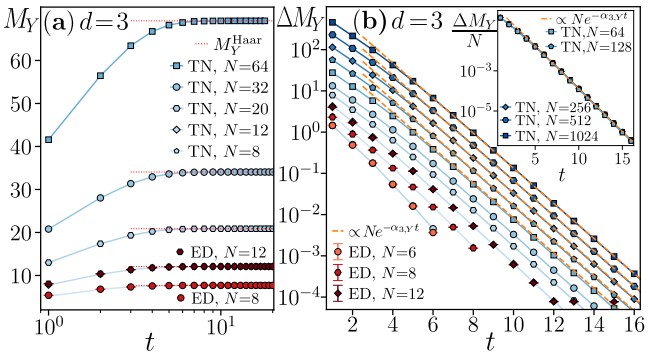

**Fig. 4 | Dynamics of the generalized stabilizer entropy $M_Y$ for qutrits circuits. a** The saturation of $M_Y$ to $M_Y^{\text{Haar}}$ occurs similarly to the qubit case. **b** The difference $\Delta M_Y = M_Y^{\text{Haar}} - M_Y$ follows $\Delta M_Y \propto N e^{-\alpha_{3,Y}t}$ with $\alpha_{3,Y} = 0.98(2)$ at $t \gtrsim 5$; see the inset. The quenched $\bar{M}_Y$ and annealed $\overline{M}_Y$ averages approach each other with the increase of $N$.

## Discussion

The brick-wall Haar random quantum circuits combine principles of locality and unitarity of time dynamics, serving as minimal models for ergodic quantum many-body systems[41]. Random circuits provide insights into the entanglement growth[27,42,43], the properties of operator spreading[44–46] and spectral correlations[47–49]. In particular, the magic resources in eigenstates of ergodic many-body systems share

properties with states of deep random circuits[37]. Hence, we conjecture that the universal features of the GSEs growth (10), i.e., the exponential relaxation to the saturation value at times $t_{\text{sat}}^{\text{mag}} \propto \ln N$ characterize the spreading of GSEs in chaotic many-body systems. Our conjecture is supported by the following observation based on the Suzuki-Trotter decomposition[50]. Consider quantum dynamics generated by a local ergodic quantum Hamiltonian $H = \sum_j H_{j,j+1}$. The Suzuki-Trotter formula is a key element of algorithms computing time dynamics of many-body system, e.g., the time-evolving block decimation[51], and allows for the following approximation of the evolution operator

$$e^{-i\Delta t H} \approx \prod_{k=1}^{N/2} e^{-i\Delta t H_{2k-1,2k}} \prod_{k=1}^{N/2-1} e^{-i\Delta t H_{2k,2k+1}},\qquad(11)$$

valid for sufficiently small $\Delta t$. Eq. (11) reproduces the structure of the brick-wall quantum circuit, with unitary gates given by $U_{k,k+1}^H = e^{-i\Delta t H_{k,k+1}}$. The gates $U_{k,k+1}^H$ for generic ergodic many-body systems do not belong to the Clifford group, and their action increases the nonstabilizerness of the state of the system. Hence, each layer of the circuit defined by (11) contains extensively many gates that increase magic resources, similar to the brick-wall Haar random quantum circuits considered in this work.

The uncovered phenomenology of the GSEs parallels, as we argue in Supplemental Note 6, the growth of mana[8] under the dynamics of random quantum circuits, even though mana is a magic state resource theory monotone not belonging to the family of GSEs. The behavior of these nonstabilizerness measures is reminiscent of time evolution of participation entropy[52], which characterizes the spread of many-body states in a selected basis of the Hilbert space and saturates at times $t_{\text{sat}}^{(\text{pe})} \propto \ln N$[53]. The latter is tied to anticoncentration of the state of logarithmically deep random circuits[54,55], which is a necessary assumption of the formal proofs underlying quantum advantage. The rapid growth of nonstabilizerness is in a stark contrast with the ballistic increase of entanglement entropy under ergodic many-body dynamics[27,56], resulting in a saturation timescale $t_{\text{sat}}^{\text{ent}} \propto N$, linear in system size. At a formal level, the difference arises due to the disparity in boundary conditions at the top layer of the circuit corresponding to the GSEs (9) and the entanglement entropy[42] calculations. Physically, the time required to entangle two distant regions by local quantum dynamics scales linearly with the separation between the regions, consistent with the scaling of $t_{\text{sat}}^{\text{ent}}$. In contrast, the GSEs capture global properties of the state, and already time $t_{\text{sat}}^{\text{ent}} \propto \ln N$ is sufficient for nonstabilizerness to equilibrate even though entanglement between the most distant qudits has not yet been generated.

In summary, in this paper we have explored the dynamics of magic resources focusing on brick-wall Haar random unitary circuits. To that end, we considered the GSEs $M_W$ as scalable measures of nonstabilizerness, which include, but are not limited to, the SRE. Our investigations reveal that magic resources are rapidly generated by the dynamics of random unitary circuits and saturate at relatively short times which scale logarithmically with the system size. The revealed behavior of $M_W$ aligns with the log-depth anticoncentration of random quantum circuits[54] and matches the phenomenology of Hilbert space delocalization under random circuits[53]. The GSE spreading remains qualitatively different from the ballistic growth of entanglement entropy in ergodic many-body systems. Since the random circuits constitute a minimal model of local unitary dynamics, we expect a similar phenomenology of magic state resources evolution to arise in generic ergodic many-body systems.

Understanding how the phenomenology of nonstabilizerness generation changes when the ergodicity is broken due to, e.g., many-body localization[57,58] or quantum scars[59], is an open question. Steps in that direction were already taken for integrable systems[26,60,61], and doped Clifford circuits[62] in which the generation of the magic resources is slower due to sparseness of beyond-Clifford operations,

cf. the Supplementary Note 6. The asymmetry between the generation of magic resources and entanglement by local dynamics provides a new perspective onto the relation of entanglement and magic phase transitions[63-66]. The framework based on the algebraic structure of the Clifford group, which yielded the GSEs, hosts more examples of magic measures with potential for better characterization of magic in many qudit systems. We leave these problems open for further research.

# Methods

## Numerical simulations

We employ two complementary numerical approaches: (i) exact circuit simulation and (ii) tensor network contraction for the annealed average of GSE. Together, they provide crucial insights into magic spreading in random quantum circuits. Exact simulation reveals the self-averaging properties of the SRE and GSE, but is limited to small system sizes. Exploiting this fact, we use tensor network contraction to compute annealed averages, which accurately approximate quenched averages, as we argue in the following.

**Self-averaging.** Exact numerical simulation of quantum circuits' dynamics involves generating the computational basis $\mathcal{B} = \{|\mathbf{x}_1\mathbf{x}_2\cdots\mathbf{x}_N\rangle \,|\, \mathbf{x}_j \in \mathbb{Z}_d\}$ and expressing the state $|\Psi_t\rangle$ as a superposition of the computational basis states. Then, the action of two-body Haar-random gates on states in $\mathcal{H}_{N,d}$ reduces to sparse matrix-vector multiplication. The first limiting factor of the exact simulation is the exponential growth of $|\mathcal{B}| = d^N$ with the system size. A more severe constraint arises from the need to evaluate $|\mathcal{P}_N(d)| = d^{2N}$ Pauli string expectation values in the time-evolved state $|\Psi_t\rangle$ to compute the GSEs of interest, cf. Main Text. We developed and employed an efficient numerical algorithm, allowing these exact calculations to system sizes up to $N = 22$ qubits and $N = 12$ qutrits.

We employ the exact numerical simulation to calculate the quenched average of the GSEs $\overline{M}_W = -\mathbb{E}[\ln(\zeta_W(|\Psi_t\rangle))]$ and the annealed average $\tilde{M}_W = -\ln[\mathbb{E}[\zeta_W(|\Psi_t\rangle)]]$, where the circuit average involves 1000 realizations unless otherwise specified. Additional details are presented in the Supplementary Note 6.

Focusing on the difference $\delta M_W(t) = |\overline{M}_W(t) - \tilde{M}_W(t)|$ between the quenched and annealed averages of SRE and GSE finding that

$$\delta M_2(t) = a_{t,2}N + b_t, \tag{12}$$

where $a_{t,2}$ and $b_{t,2}$ are constant at fixed time $t$. This behavior characterizes $\delta M_2(t)$ both for qubits ($d = 2$) and qutrits ($d = 3$). We note that analogous behavior holds for $M_Y(t)$. After an initial transient at small circuit depths $t$, the coefficients $a_{t,W}$ decrease exponentially over time for all relevant operators $W$ in both qubit and qutrit systems

$$a_{t,W} = a_W e^{-\beta_{d,W}t}, \tag{13}$$

where $\beta_{d,W}$ is a constant dependent on the on-site Hilbert space dimension $d$ and on $W$. For qubits, we find $\beta_{2,2} = 0.83(3)$ while for qutrits $\beta_{3,2} = \beta_{3,Y} = 1.97(5)$. These values of $\beta_{d,W}$ confirm that the error made when the quenched averages are interchanged with the annealed averages is negligible for sufficiently large $N$ and $t$. Indeed, the exponential decay of $\Delta M_W(t)$, as described in Eq. (10), occurs at a rate $\alpha_{2,2} = 0.43(3)$ for qubits and $\alpha_{3,2} = 1.05(3)$ for qutrits for $M_2$, while, for qutrits, $\Delta M_Y(t)$ decays with a rate of $\alpha_{3,Y} = 0.98(2)$. The rates $\beta_{d,W}$ are significantly (approximately twice) larger than the corresponding rates $\alpha_{d,W}$. Hence, the relative errors committed when the quenched and annealed averages are interchanged decays exponentially in time as

$$\delta M_W(t)/\Delta M_W(t) \propto e^{-(\beta_{d,W} - \alpha_{d,W})t}, \tag{14}$$

up to a sub-leading in system size term $O(1/N)$. For any $t \geq 0$, the relative error is smaller than 3% in all the considered cases and a clear

exponential decay of the relative error is observed at $t \gtrsim 5$–10. All the details are included in the Supplementary Note 6.

The above scaling analysis demonstrates that approximating quenched averages with the annealed averages in the dynamics of brick-wall quantum circuits is justified at any time $t$ and that this approximation improves exponentially with $t$ as shown by (14). Moreover, our numerical results indicate that the linear scaling (12) is robust and persists at any system size $N$. These two trends show that our main conclusion about the magic resources growth in random quantum circuits, i.e. the logarithm depth saturation of magic resources, $t_{\text{sat}}^{\text{mag}} \propto \ln(N)$, is accurate in the scaling limit of large system size $N$.

**Tensor network contractions.** Our tensor network approach aims at an efficient evaluation of the tensor network contraction (9) which yields the annealed average $\tilde{M}_W(t)$. To that end, we interpret (9) as a non-unitary time evolution of a state of $N$ effective "spins" with on-site Hilbert space dimension $q_{\text{eff}} = k!$ growing factorially with the number $k$ of replicas. The non-unitary time evolution is followed by the contraction of the obtained state with the last layer (8).

The state of the effective spins is expressed as a matrix product state (MPS)

$$|\Pi\rangle = \sum_{\tau_1, \ldots, \tau_N \in S_k} A_{[1]}^{\tau_1} A_{[2]}^{\tau_2} \cdots A_{[N]}^{\tau_N} |\tau_1, \ldots, \tau_N\rangle, \tag{15}$$

where $|\tau_1, \ldots, \tau_N\rangle = |\tau_1\rangle \otimes |\tau_2\rangle \cdots \otimes |\tau_N\rangle$ is the product state of representations of permutations $\tau_i \in S_k$ while $A_{[i]}^{\tau_i}$ are $\chi \times \chi$ matrices for any $i = 2, \ldots, N-1$, $A_{[1]}^{\tau_1}$ ($A_{[N]}^{\tau_N}$) are $1 \times \chi$ ($\chi \times 1$) matrices and $\chi$ is the bond dimension, which needs to be contrasted with the on-site local Hilbert dimension $q_{\text{eff}}$. At $t = 1$, the state of the system $|\Pi_1\rangle$ is the product state (7).

The time evolution consists of contraction of the state of the system $|\Pi_t\rangle$ with subsequent layers of the tensors $\mathcal{T}_{i,i+1}^{(k)}$ defined by (6). The contraction of $\mathcal{T}_{i,i+1}^{(k)}$ with the matrices $A_{[i], a_i, a_{i+1}}^{\tau_i}$, $A_{[i+1], a_{i+1}, a_{i+2}}^{\tau_{i+1}}$ (where $a_i$, $a_{i+1}$, $a_{i+2}$ denote the matrix indices of $A_{[i]}$, $A_{[i+1]}$) results in a tensor $\mathcal{T}_{a_i', a_{i+2}'}^{\tau_i, \tau_{i+1}}$. Reshaping the tensor to $\mathcal{T}_{(\tau_i, a_i), (\tau_{i+1}, a_{i+2})}$ results in a matrix of dimension $(q_{\text{eff}}\chi) \times (q_{\text{eff}}\chi)$, which is expressed back as a product of two matrices $A_{[i]}'$, $A_{[i+1]}'$ via the standard singular value decomposition (SVD). Notably, the dimension of the matrix $\mathcal{T}_{(\tau_i, a_i), (\tau_{i+1}, a_{i+2})}$ is increased by a factor $q_{\text{eff}}$ with respect to the bond dimension $\chi$. The complexity of SVD scales as $(q_{\text{eff}}\chi)^3$ which is the main problem hindering our calculations for qubits ($d = 2$), since the on-site Hilbert space dimension is $q_{\text{eff}} = 4! = 24$. This fact was one of our motivations for introducing the GSE $M_Y$ and considering qutrits, which results in a much smaller on-site Hilbert space dimension $q_{\text{eff}} = 3! = 6$ and a significantly simpler computation of the tensor network contraction. Finally, for qubits ($d = 2$), as noted in ref. 40, the effective on-site Hilbert space dimension can be reduced from $q_{\text{eff}} = k!$ to $q_{\text{eff}}' = C_k$, where $C_k$ is the Catalan number. This enables us to calculate $\tilde{M}_2(t)$ for qubits by considering model with $q_{\text{eff}}' = C_4 = 14$ dimensional on-site Hilbert space. To find the mapping between the tensors of (9) living $q_{\text{eff}} = k!$ dimensional space to the $q_{\text{eff}}' = 14$ subspace, we employed the single qubit Haar-random gate which, upon averaging, corresponds to the projector onto the 14-dimensional invariant subspace.

The results presented in the Main Text are converged with the bond dimension $\chi$ of the MPS (15). The convergence occurs once $\chi \approx q_{\text{eff}}^2$, see Supplementary Note 6 for additional details.

## Results beyond generalized stabilizer entropies: mana

In the Main Text, we considered the growth of GSEs under dynamics of Haar random brick-wall circuits acting on systems of qubits and qutrits finding the exponential relaxation of these nonstabilizerness measures towards their long time values (10). Here, we argue that analogous phenomenology is shared by mana, a nonstabilizerness monotone defined for odd prime $d$ as the negativity of the Wigner representation of the state $\rho$[8].

The phase-space point operators $A_\mathbf{r}$ are defined in terms of the Pauli strings $P_\mathbf{r} = \prod_{l=1}^{N} X_l^{r_l^x} Z_l^{r_l^z}$ (where $\mathbf{r} \equiv (r_1^x, r_1^z, \ldots, r_N^x, r_N^z) \in \mathbb{Z}_d^{2N}$) as

$$A_\mathbf{0} = \frac{1}{d^N} \sum_{\mathbf{r} \in \mathbb{Z}_d^{2N}} P_\mathbf{r}, \quad A_\mathbf{r} = P_\mathbf{r} A_\mathbf{0} P_\mathbf{r}^\dagger. \tag{16}$$

The phase-space point operators are orthogonal, $\mathrm{tr}(A_\mathbf{r} A_{\mathbf{r}'}) = d^N \delta_{\mathbf{r}, \mathbf{r}'}$, enabling to represent the state as

$$\rho = \sum_{\mathbf{r} \in \mathbb{Z}_d^{2N}} \frac{1}{d^N} \mathrm{tr}(A_\mathbf{r} \rho) A_\mathbf{r}. \tag{17}$$

The coefficients of the expansion define the discrete Wigner function $W_\rho(\mathbf{r}) = \frac{1}{d^N} \mathrm{tr}(A_\mathbf{r} \rho)$[9]. Mana $\mathcal{M}$ is defined as the negativity of the Wigner function

$$\mathcal{M} = \ln\left(\sum_\mathbf{u} |W_\rho(\mathbf{u})|\right). \tag{18}$$

Mana offers insights into magic state resources as it is a strong nonstabilizerness monotone both for pure and mixed states.

Performing exact numerical simulations of Haar-random brickwall circuits for qutrits and calculating mana, we observe that $\mathcal{M} \propto N$ already at $t = 1$, after a single layer of the circuit. Additionally, $\mathcal{M}$ quickly saturates with $t$ to the value $\mathcal{M}^{\mathrm{Haar}}$ of the Haar-random state of $N$ qutrits. The difference $\Delta\mathcal{M} = \mathcal{M}^{\mathrm{Haar}} - \mathcal{M}(t)$ follows an exponential decay with circuit depth $t$, $\Delta\mathcal{M} = b_n e^{-\alpha t}$, with the prefactor $b_n$ increases linearly with $N$ and $\alpha$ converging to a constant with increase of $N$, see the Supplementary Note 6 for further details. These numerical results suggest that $\Delta\mathcal{M} \propto N e^{-\alpha t}$, analogously to the GSEs (10). This demonstrates the universality of the uncovered phenomenology of magic spreading among different measures of nonstabilizerness.

### Stabilizer Rényi entropies are magic monotone for qudits

A key contribution of this paper is proving that stabilizer Rényi entropies are magic monotone under general stabilizer protocols for any qudit with dimension $d$ prime. Following ref. 12, we require only to bound the expectation value $P_\alpha(\Psi) \equiv d^{-N} \sum_{P \in \mathcal{P}_N} \langle \Psi | P | \Psi \rangle^{2\alpha}$ for a state on $N$ qudits $|\Psi\rangle = \sum_{i=0}^{d-1} \sqrt{p_i} |i\rangle \otimes |\phi_i\rangle$ with $\sum_{i=0}^{d-1} p_i = 1$, $\{|i\rangle\}_{i=0,\ldots,d-1}$ on-site state in the computational basis, and $\{|\phi_i\rangle\}_{i=0,\ldots,d-1}$ generic states on the remaining $N-1$ qudits. In particular, we must show the following Lemma holds:

**Theorem 1.** Consider $|\Psi\rangle = \sum_{i=0}^{d-1} \sqrt{p_i} |i\rangle \otimes |\phi_i\rangle$ and $|\phi_i\rangle \in \mathbb{C}^{d\otimes(N-1)}$. For any integer $\alpha \geq 2$ it holds that

$$P_\alpha(\Psi) \leq \max_j \{P_\alpha(\phi_j)\}. \tag{19}$$

We sketch the proof of the lemma, closely following the technique in Ref. 12, and present the full details in the Supplementary Note 3 A. The key technique is to expand $P_\alpha(\Psi)$ over $\tilde{P} \otimes P$ with $P \in \mathcal{P}_{N-1}$ and $\tilde{P} \in \mathcal{P}_1$

$$P_\alpha(\Psi) = \sum_{\substack{P \in \mathcal{P}_{N-1} \\ \tilde{P} \in \mathcal{P}_1}} \frac{1}{d^N} \left| \sum_{i,j=0}^{d-1} \langle i | \tilde{P} | j \rangle \langle \phi_i | P | \phi_j \rangle \sqrt{p_i p_j} \right|^{2\alpha}, \tag{20}$$

and to explicitly the sum over $\tilde{P}$ using one qudit Pauli operator. After straightforward but lengthly algebra based on the binomial expansion and repeated applications of Cauchy-Schwarz and Hölder inequalities, we obtain

$$P_\alpha(\Psi) \leq \max_m \{P_\alpha(\phi_m)\} \sum_{|\mathbf{i}|, |\mathbf{j}| = \alpha} \binom{\alpha}{\mathbf{i}} \binom{\alpha}{\mathbf{j}} \tilde{\delta}_{\mathbf{i}, \mathbf{j}} F_{\mathbf{i}, \mathbf{j}}(\mathbf{p}), \tag{21}$$

where $|\mathbf{i}| = \sum_{m=0}^{d-1} i_m$, $F_{\mathbf{i}, \mathbf{j}}(\mathbf{p}) \equiv \sum_{a=0}^{d-1} \prod_{m=0}^{d-1} (p_{j\oplus a} p_j)^{j_m + i_m}$, $\binom{\alpha}{\mathbf{i}} = \alpha!/(i_0! i_1! \ldots i_{d-1}!)$ is the multinomial coefficient, and $\tilde{\delta}_{\mathbf{i}, \mathbf{j}} \equiv \delta_{\sum_m m(j_m - i_m) = 0 \bmod d}$ is enforced by the phases of the $\tilde{P}$ operators. Nevertheless, since $F_{\mathbf{i}, \mathbf{j}}(\mathbf{p}) \leq d^{1-2\alpha}$ for all $\mathbf{i}$ and $\mathbf{j}$, using standard properties of the multinomial coefficient, it follows that $P_\alpha(\Psi) \leq \max_m \{P_\alpha(\phi_m)\}$ as required.

## Data availability

The data generated in this study have been deposited in the zenodo public folder[67].

## Code availability

The code generated in this study have been deposited in the zenodo public folder[67]. Part of the code employs the open-source ITENSOR library[39].

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

## Acknowledgements

We are grateful to D. Gross, L. Leone, and A. Hamma for enlightening discussions, R. Fazio for valuable collaborations and comments, and M.

Lewenstein for collaborations on related topics. X.T. acknowledges DFG under Germany's Excellence Strategy - Cluster of Excellence Matter and Light for Quantum Computing (ML4Q) EXC 2004/1 - 390534769, and DFG Collaborative Research Center (CRC) 183 Project No. 277101999 - project B01. E.T. was supported by the MIUR Programme FARE (MEPH), by QUANTERA DYNAMITE PCI2022-132919, and by the EU-Flagship programme Pasquans2. E.T. acknowledge the CINECA award under the ISCRA initiative, for the availability of high-performance computing resources and support. P.S. acknowledges support from the European Research Council AdG NOQIA; MCIN/AEI (PGC2018-0910.13039/501100011033, CEX2019-000910-S/10.13039/501100011033; Plan National STAMEENA PID2022-139099NB; Ministry for Digital Transformation and of Civil Service of the Spanish Government through the QUANTUM ENIA project call - Quantum Spain project, and by the European Union through the Recovery, Transformation and Resilience Plan - NextGenerationEU within the framework of the Digital Spain 2026 Agenda; Fundació Cellex; Fundació Mir-Puig; the computing resources at Urederra and technical support provided by NASERTIC (RES-FI-2024-1-0043). P.S. acknowledges fellowship within the -"Generación D" initiative, Red.es, Ministerio para la Transformación Digital y de la Función Pública, for talent atraction (C005/24-ED CV1), funded by the European Union NextGenerationEU funds, through PRTR. Views and opinions expressed are however those of the author(s) only and do not necessarily reflect those of the European Union, or any other granting authority.

## Author contributions

X.T. developed the mathematical formalism and the replica tensor network computation. P.S. and E.T. contributed to the numerical simulations. P.S. developed the numerical exact diagonalization and optimized the replica tensor network approach. X.T. and P.S. equally contributed to the writing. All authors contributed to the analysis.

## Funding

## Competing interests

The authors declare no competing interests.
