## [Transparent Peer Review file · Nature Communications]

Magic spreading in random quantum circuits

Corresponding Author: Dr Xhek Turkeshi

Version 0:

Reviewer comments:

Reviewer #1

(Remarks to the Author)

The manuscript "Magic Spreading in Random Quantum Circuits" introduces a new measure of magic called CSS entropy and then analyzes the magic spreading due to random quantum circuits.

I carefully read the manuscript and do not believe it is suitable for publication in Nature Communications. I would instead suggest publication in more technical journals. While the manuscript is well-written and clear in its intent, I fail to see the significance of the results. The newly introduced quantity does not appear to add anything new to what is already known in the literature. Additionally, the analysis of random quantum circuits employs common and well-known techniques, such as Weingarten calculus and tensor network methods, but it does not present anything of significant conceptual interest.

Major Comments

1. Monotonicity: The manuscript introduces a new measure, called CSS entropy. While some necessary properties for it to be a magic monotone are demonstrated, there is no mention of the monotonicity of the quantity, which is fundamental for defining a measure of magic. It is clear that for $d = 2$, the proof follows from the proof made for the Stabilizer Rényi entropy, yet no progress is made toward proving it for $d \neq 2$, where the generalization would be of more significant interest.
2. Operational Meaning: The authors fail to introduce the operational meaning of these quantities and why the CSS entropy should be considered in favor of other measurable and computable measures of magic, such as mana in the case of odd dimensions or generalizations of stabilizer Rényi entropy (even though it has not been shown to be a monotone) for any dimensions. The only novelty of this measure seems to be the recognition that, not only for qubits but also for qudits, one can use the projectors on the stabilizer subspace to test magic.
3. The Average $\langle Y \rangle$: While I understand the choice of using the average $\langle Y \rangle$ after the saturation time, the approximation fails to be trustworthy before this point, as evident from your models. The relative error appears to be at least $O(1)$, and the approximation only becomes valid when the circuits can be approximated by a random unitary sampled from the Haar measure.
4. Saturation Time: On the saturation time, the authors show that contrary to what happens for the entanglement, the saturation time for the entanglement is $t = O(\log n)$. While in principle this could appear surprising, it is important to remark that contrary to what happens for the entanglement, where from an area law, one steps to a volume law of entanglement, here we are considering the case in which magic is already extensive, meaning that there is less room for increasing, and such increase is in general connected to the entanglement.

Minor Comments and Questions:

1. I think the term "entropy" is somewhat misleading since the quantity is the logarithm of an expectation value, and it only becomes a probability distribution for $d = 2$.
2. Why are two different quantities used in the second figure and the insets of Figs. (2) and (3)? I think that using $\Delta Y/N$ can be quite misleading when making a direct comparison of the curves.

Reviewer #2

(Remarks to the Author)

In the paper the authors study the behavior of magic in random quantum circuits. Using replica methods combined with tensor network numerics, they are able to systematically study the behavior of magic in local random circuits acting on qubits

and qutrits up to hundreds or thousands of sites. The result that magic saturates in log-depth even in 1D circuits is physically reasonable and a clear intuition is provided for this behavior which sharply contrasts with entanglement entropy. The numerical results are also impressive as the solution of the replica theory is quite challenging for more than 3 replicas.

One comment on the analysis of qubits is that the effective dimension for 4 replicas is actually less than 24 because and gets reduced to 14. This point could be used to improve the numerical results. In addition, there are many symmetries in the replica model that should allow further reductions in the local dimension. I believe the results for qubits could be improved substantially by taking advantage of these effects.

Overall, I find the paper timely, accurate and clearly presented and recommend publication in Nature Communications.

Version 1:

Reviewer comments:

Reviewer #1

(Remarks to the Author)

I have reviewed the revised materials and appreciate the authors' thorough efforts in addressing the referee's comments. Most suggestions have been effectively incorporated, with valuable additions that enhance the manuscript. Overall, its quality has significantly improved.

I carefully considered the revised materials in assessing this work's suitability for Nature Communications. From the outset, I found the topic and many of the results conceptually interesting. The authors have done an excellent job exploring and presenting their findings, especially after this revision.

However, my critical comments on the work's strength and impact remain, particularly in comparison to the stringent standards of Nature Communications. While I acknowledge the authors' effort in analyzing the spreading of magic in ergodic systems, I do not find the results sufficiently significant for this journal. That said, I believe the paper could be a valuable contribution to other, more specialized journals. Below are some additional comments.

- I suggest highlighting the technical result on the monotonicity of the stabilizer Rényi entropies, as the techniques may be of interest to the community.

- There is a missing citation in the Conclusions near the sentence: "Steps in that direction were already taken for integrable systems."

Reviewer #2

(Remarks to the Author)

The authors have made several nice changes to the manuscript in response to the referee comments. I recommend the paper for publication in Nature Communication.

Reply to Reviewer 1

The manuscript "Magic Spreading in Random Quantum Circuits" introduces a new measure of magic called CSS entropy and then analyzes the magic spreading due to random quantum circuits. I carefully read the manuscript and do not believe it is suitable for publication in Nature Communications. I would instead suggest publication in more technical journals.

We thank the Referee for their reading of the manuscript, and their comments.

While the manuscript is well-written and clear in its intent, I fail to see the significance of the results. The newly introduced quantity does not appear to add anything new to what is already known in the literature. Additionally, the analysis of random quantum circuits employs common and well-known techniques, such as Weingarten calculus and tensor network methods, but it does not present anything of significant conceptual interest.

We thank the Reviewer for these remarks, which identified several points where the relevance of our work was not sufficiently highlighted. We believe that our revision has addressed these issues by presenting in a more transparent way (i) the scope of our work and the conceptual question it tackles and resolves, (ii) the significance of our work and how it goes substantially beyond the state-of-the-art, (iii) the technical improvements obtained in this manuscript. (Below, we use naming conventions and denotations consistent with the revised version of our manuscript.)

- **Conceptual interest:** We tackle and solve an open problem in quantum many-body dynamics: how magic spreads in ergodic systems. This question was previously unresolved, and the only paper that addressed this question earlier tied the dynamics of magic to that of entanglement entropy in an integrable system for $N \leq 16$. We demonstrated clearly that this is not the case for generic ergodic systems, instead linking the phenomenology magic spreading to that of anticoncentration. The generality of quantum circuits, and their success in capturing the qualitative, coarse-grained features of many-body dynamics suggest that our results are generically valid for any ergodic many-body systems.
- **Scope of the introduced nonstabilizerness measures:** The scope of the introduced quantities, referred to as generalized stabilizer entropies (**GSEs**), in the revised manuscript, is twofold. On the one hand, GSEs show that any non-trivial (non-permutation) element of the Clifford commutant induces a well-behaved measure of magic. The second is that the GSE allows us to obtain numerical results for up to thousands of sites for qutrits, as we need only three copies of the system to obtain M_Y . As our numerical results demonstrate decisively, time evolution of M_Y shares the phenomenology of other measures of nonstabilizerness such as stabilizer Rényi entropies and mana.
- **Technical relevance:** Implementing replica tensor network beyond two copies of the system is a major computational challenge due to the dimension of the effective spins, that grows factorially as $d_{\text{eff}} \sim k!$ with the replica number k . While we are aware of studies for $k = 2$ replicas, to the best of our knowledge our work is the first study for $k = 4$ replicas. This, as acknowledged by Reviewer 2, is an impressive computational step forward, which enabled us to understand the magic spreading in local many-body dynamics and opens up new pathways for understanding the time evolution of random quantum circuits.

We hope these points clarify the significance of our work. Below, we have addressed all the questions and comments of the Referee. At the end of this file, we summarize the main changes in the revised manuscript.

1 **Monotonicity:** The manuscript introduces a new measure, called CSS entropy. While some necessary properties for it to be a magic monotone are demonstrated, there is no mention of the monotonicity of the quantity, which is fundamental for defining a measure of magic. It is clear that for $d = 2$, the proof follows from the proof made for the Stabilizer Rényi entropy, yet no progress is made toward proving it for $d \neq 2$, where the generalization would be of more significant interest.

We thank the Referee for this question and would like to answer it by discussing the scope of our findings and the monotonicity of the considered non-stabilizerness measures.

Our manuscript focuses on the spreading of magic resources under many-body dynamics. This problem is important and timely since nonstabilizerness is a resource necessary for reaching quantum advantage. Additionally, nonstabilizerness is of fundamental interest as it provides a new perspective on the complexity of quantum states. Our work significantly advances the understanding of the nonstabilizerness growth in many-body dynamics by providing the first large-scale results for the time evolution of non-stabilizerness measures. Accessing the extensive system sizes $N \gg 1$ is essential for formulating the conclusions for the coarse-grained growth of magic resources and, in particular, for understanding the time scales at which nonstabilizerness saturates. For phenomenological predictions on the growth of magic under unitary dynamics, the monotonicity of specific nonstabilizerness measures under general stabilizer protocols (including computational basis measurements) is not a fundamental concern.

Prompted by the Referee, we analyzed the monotonicity of the GSE M_Y [dubbed CSS entropy in the previous version of our manuscript] for qutrits. A real-valued function $f(\rho)$ is a stabilizer monotone if, for any state ρ and a stabilizer protocol $\mathcal{E} : \rho \rightarrow \mathcal{E}(\rho)$,

$$f(\mathcal{E}(\rho)) \leq f(\rho). \quad (1)$$

To study the monotonicity of GSE M_Y under computational basis measurements, we consider (without loss of generality) a measurement of the first qudit and expand the pure state $|\Psi\rangle$ of N qudits as

$$|\Psi\rangle = \sum_{i=0}^{d-1} \sqrt{p_i} |i\rangle \otimes |\phi_i\rangle, \quad (2)$$

where $\{|i\rangle\}_{i=0}^{d-1}$ are the computational basis states, $\sum_{i=0}^{d-1} p_i = 1$, and $|\phi_i\rangle$ are the states of the remaining $N - 1$ qudits. Analyzing the behavior of the M_Y under measurements in the computational basis, we realized that the GSEs, in general, *are not* monotones. Already for $N = 2$ and qutrits, we found a counter-example $|\Psi_2\rangle = \sum_{i,j=0}^2 a_{i,j} |i\rangle \otimes |j\rangle$, where $a_{00} = 0.04899 + 0.29503i$, $a_{10} = 0.11566 + 0.04942i$, $a_{20} = 0.24715 + 0.34531i$, $a_{01} = 0.37355 + 0.23511i$, $a_{11} = 0.15912 + 0.33928i$, $a_{21} = 0.33144 + 0.08953i$, $a_{02} = 0.10273 + 0.08267i$, $a_{12} = 0.43260 + 0.22725i$, $a_{22} = 0.02948 + 0.06498i$. For this state, we have $M_Y(|\Psi_2\rangle\langle\Psi_2|) \approx 0.7837$, which is *smaller* than the GSE values for each of the states obtained after the measurement of the first qutrits in computational basis: $M_Y(|\phi_1\rangle\langle\phi_1|) \approx 1.0004$, $M_Y(|\phi_2\rangle\langle\phi_2|) \approx 1.0408$, $M_Y(|\phi_3\rangle\langle\phi_3|) \approx 1.4456$. This counter-example shows that not all nonstabilizerness measures induced by the elements of the *intrinsic* Clifford commutant are monotones.

In contrast, the monotonicity of M_2 for the magic-state resource theory was proven in [Phys. Rev. A **110**, L040403 (2024)] for qubits. In the Supplemental Material of the revised manuscript, we extend the proof for arbitrary $d \geq 2$ by showing that $M_2(|\Psi\rangle\langle\Psi|) \geq \min_i(M_2(|\phi_i\rangle\langle\phi_i|))$ for any N . Thus, while the GSE M_Y is not a monotone, the SRE M_2 is a monotone for systems of qutrits. Nevertheless, the GSE M_Y behavior for qutrits (Fig. 4 of the revised manuscript) closely follows the growth and saturation of SRE M_2 for qutrits presented in Fig. 3 of the revised manuscript. Thus, both M_Y and M_2 lead to the same conclusion about the time evolution of nonstabilizerness, independently of their monotonicity under the general stabilizer protocols. Moreover, in the section Methods of the revised manuscript, we argue the same phenomenology is shared by mana [New J. Phys. **14**, 113011 (2012)], the renowned magic-state resource theory monotone. Unfortunately, mana does not belong to the family of GSEs and does not have

a clear representation within the replica formalism. This limits our investigations to brick-wall circuits acting on chains comprising at most $N = 14$ qutrits.

2. **Operational Meaning:** The authors fail to introduce the operational meaning of these quantities and why the CSS entropy should be considered in favor of other measurable and computable measures of magic, such as mana in the case of odd dimensions or generalizations of stabilizer Rényi entropy (even though it has not been shown to be a monotone) for any dimensions. The only novelty of this measure seems to be the recognition that, not only for qubits but also for qudits, one can use the projectors on the stabilizer subspace to test magic.

We would like to emphasize that we never claim that the CSS entropy (or GSE M_W in the naming convention of the revised manuscript) should be used instead of other measurable and computable measures of magic. The point of our work is that *various measures of nonstabilizerness* increase and saturate following *the same phenomenology*, identified in our work. To highlight this point, in the methods of the revised manuscript, we provide numerical evidence showing that mana (which does not belong to the family of GSEs) behaves in an analogous manner, saturating at timescales scaling logarithmically with N .

Given the focus of our manuscript, i.e., understanding the growth of magic resources under local many-body dynamics, the operational meaning of the GSE is of secondary importance for us. The reason for which the GSE is introduced in our manuscript is that it can be calculated for qutrits with the use of 3-replicas only (we note that such a measure does not exist for qubits for which Clifford group forms a 3-design [Phys. Rev. A 96, 062336 (2017)]). This provides a radical simplification of the tensor network contractions, reducing the effective on-site Hilbert space dimension from $d_{\text{eff}} = 4! = 24$ required for SRE, to $d_{\text{eff}} = 3! = 6$. This point is discussed in details in Methods of the manuscript. Importantly, even though the GSE is not a monotone, its growth follows the same phenomenology as the growth of SRE which we have shown to be a monotone for qudits.

3. **The Average \tilde{Y} :** While I understand the choice of using the average \tilde{Y} after the saturation time, the approximation fails to be trustworthy before this point, as evident from your models. The relative error appears to be at least $O(1)$, and the approximation only becomes valid when the circuits can be approximated by a random unitary sampled from the Haar measure.

We thank the Referee for bringing up the question of the relation between the quenched and annealed averages of GSEs M_W . This relation is critical for understanding the validity of our results. In the following, we clarify that the quenched, \overline{M}_W , and annealed, \tilde{M}_W , averages can be used interchangeably to quantify non-stabilizerness at any circuit depth. To that end, we present a detailed scaling analysis of the differences between \overline{M}_W and \tilde{M}_W under the dynamics of Haar random brick-wall circuits.

We computed the state vectors corresponding to the time-evolved states $|\Psi_t\rangle$ and performed exact numerical calculation of the quenched and annealed averages, $\overline{M}_W \equiv \mathbb{E}[M_W(|\Psi_t\rangle)] = -\mathbb{E}[\log[\zeta_W(|\Psi_t\rangle)]]$ and $\tilde{M}_W \equiv -\log[\mathbb{E}[\zeta_W(|\Psi_t\rangle)]]$. To conduct the comparison in the widest range of system sizes possible, we pushed the numerics to systems of $N = 22$ qubits ($d = 2$) and $N = 12$ for qutrits ($d = 3$). Performing these numerical calculations required over 1.5 millions of CPU hours. Additionally, for qubits, we performed tensor network calculations of $|\Psi_t\rangle$ and calculated the quenched and annealed averages by expressing the matrix product state representing $|\Psi_t\rangle$ in the Pauli basis [Phys. Rev. Lett. 133, 010601 (2024)], which enabled us to reach $N = 32$ for $t < 5$.

Our comparison of the quenched and annealed averages is shown in Fig. 1. We observe that the quenched averages, $\overline{M}_W(t)$, denoted by the symbols, accurately reproduce the behavior of the annealed averages $\tilde{M}_W(t)$ across all the system sizes N available to our exact calculations. This conclusion holds both at short times, $t \in [1, 10]$, as visible in Fig. 1(a),(b),(c), as well as at longer times, $t \geq 10$, at which the

Figure 1: Quenched (symbols) and annealed (solid lines) averages of SRE, $M_2(t)$ and GSE, M_Y , for systems of N qubits (a), and qutrits (b,c), as function of circuit depth t . The deviations $\Delta M_W(t)$ from the long-time saturation values are shown in (d, e, f). The quenched and annealed averages are very good approximations of each other at all times t .

quenched and annealed averages of $\Delta M_W(t)$ faithfully follow each other, as shown in Fig. 1(d),(e),(f). The results are fully analogous for the SRE $M_2(t)$ for systems of qubits and qutrits as well as for the GSE M_Y for qutrits.

The above comparison confirms the validity of using the quenched and annealed averages of non-stabilizerness measures interchangeably to quantify the dynamics of Haar random brick-wall circuits. To understand better the relation between the quenched and annealed averages, we analyze the system size scaling of the difference $\delta M_W(t) = |\overline{M}_W(t) - \tilde{M}_W(t)|$, where both $\overline{M}_W(t)$, $\tilde{M}_W(t)$ are calculated by the exact numerical simulation of the circuits. Fig. 2(a),(d) shows that $\delta M_w(t)$, at fixed t , scales linearly with N at sufficiently big system size both for qubits and qutrits (analogous behavior is observed for $\delta M_Y(t)$ and qutrits, data not shown). We parametrize this dependence as

$$\delta M_W(t) = a_{t,W}N + b_{t,W}, \quad (3)$$

where $a_{t,W}$ and $b_{t,W}$ are constant at fixed time t and operator W . The coefficient $a_{t,W}$ decreases rapidly with time (circuit depth) t , as shown in Fig. 2(b),(e). Both for qubits ($d = 2$) and qutrits ($d = 3$), after an initial transient at small circuit depths, the coefficient a_t decreases exponentially with time,

$$a_{t,W} = ae^{-\beta_{d,W}t}, \quad (4)$$

where $\beta_{d,W}$ is a constant dependent on the on-site Hilbert space dimension d and the operator W . For qubits, we find $\beta_{2,2} = 0.83(3)$ while for qutrits $\beta_{3,2} = 1.97(5)$. These values of $\beta_{d,W}$ confirm that the error made when the quenched averages are interchanged with the annealed averages is negligible for sufficiently large N and t . Indeed, recall Eq. (12) in the Main Text, which shows that

$$\Delta M_W(t) = a_{d,W}Ne^{-\alpha_{d,W}t}, \quad (5)$$

Figure 2: Difference between quenched and annealed averages of SRE, $\delta M_2(t)$, and GSE $\delta M_Y(t)$, in brick-wall Haar random circuits. For qubits, (a), and qutrits, (d), $\delta M_W(t)$ scales linearly, c.f. Eq. (3) with the system size N (for presentation purposes δM_2 is rescaled by a factor e^{t/k_d} with $k_2 = 3$ and $k_3 = 5/7$). The coefficient $a_{t,W}$ describing the leading term in this dependence, decreases exponentially with time t , consistently with Eq. 4, with exponents $\beta_{2,2} = 0.83(3)$ and $\beta_{3,2} = 1.97(5)$, see (b),(e). Consequently, the relative error, decays exponentially in time as $\delta M_W(t)/\Delta M_W(t) \propto e^{-(\beta_{d,W}-\alpha_{d,W})t}$, see (c,f).

with $\alpha_{2,2} = 0.43(3)$ for qubits and $\alpha_{3,2} = 1.05(3)$ for SRE for qutrits (and $\alpha_{3,Y} = 0.98(2)$ for $\Delta M_Y(t)$). The exponents $\beta_{d,W}$ are significantly (approximately twice) larger than the corresponding exponents $\alpha_{d,W}$. Hence, the relative error committed when the quenched and annealed averages are interchanged decays exponentially in time, $\delta M_W(t)/\Delta M_W(t) \propto e^{-(\beta_{d,W}-\alpha_{d,W})t}$ (up to a sub-leading in system size term $O(1/N)$). These relative errors are shown in Fig. 2(c),(f) for SRE for qubits and qutrits. We observe a clear exponential decay of the relative error starting at times $\approx 5 - 10$ at which the saturation of non-stabilizerness measures is observed for the system sizes considered in the Main Text. At all times, this relative error is smaller than 3% in all the considered cases. In summary, the relative error for GSEs M_W saturates below a threshold ϵ on an $O(1)$ timescale, while the average value saturates at the longer timescale $\ln(N)$.

The above scaling analysis shows the validity of our approach demonstrating that approximating quenched averages with the annealed averages of $M_W(t)$ in the dynamics of brick-wall quantum circuits is justified at *any* t , improving exponentially with time t . The exponential improvement of this approximation shows that our main result, $t_{\text{sat}}^{\text{mag}} \propto \ln(N)$ is accurate in the limit of large qubit number N , which is our main conclusion about the magic resources growth in random quantum circuits.

In the revised manuscript, we include this analysis of the relevance of annealed averages for probing magic resources growth in random quantum circuits. A direct comparison between the annealed and quenched averages of $M_W(t)$ is contained in Fig. 2,3,4 in the Main Text of the revised manuscript. In the revised Methods, we expand the scaling analysis of the error between quenched and annealed averages.

4. Saturation Time: On the saturation time, the authors show that contrary to what happens for the entanglement, the saturation time for the entanglement is $t = O(\log n)$. While in principle this could appear surprising, it is important to remark that contrary to what happens for the entanglement, where from an area law, one steps to a volume law of entanglement,

here we are considering the case in which magic is already extensive, meaning that there is less room for increasing, and such increase is in general connected to the entanglement.

The Referee correctly points out that already a single layer of the brick-wall Haar random circuits introduces an extensive value of GSE, even though we initialize the system in a product stabilizer state with vanishing GSE. This is a simple consequence of the additivity of GSE and the fact that each two-body Haar random gate increases the GSE by a non-vanishing value. From this perspective, it is indeed natural to expect that the saturation of non-stabilizerness measures may occur at time scales $t_{\text{sat}}^{\text{mag}}$ parametrically faster than linear in N times $t_{\text{sat}}^{\text{ent}}$ required for saturation of entanglement. Nevertheless, quantification of the system size dependence $t_{\text{sat}}^{\text{mag}}$ under generic local many-body dynamics *remained an important open question*, which is now *resolved by our manuscript*. We would like to emphasize that resolving this question amounts to solving a non-trivial computational problem. To understand the universal properties behavior of nonstabilizerness measures in the scaling limit of large system sizes, we needed to perform both the replica trick requiring the unprecedented 4-replica calculations resulting in effective on-site Hilbert space dimension $4! = 24$, as well as non-trivial and numerically costly exact calculations of non-stabilizerness measures required to justify the equivalence of using the quenched and annealed averages in the scaling limit.

We would like also to highlight that the phenomenology uncovered in our manuscript follows is analogous to anticoncentration, quantified by the so-called collision probability, which occurs also at time scales $\tau \sim \ln(N)$ as shown in [PRX Quantum 3, 010333 (2022)]. Hence, our result $t_{\text{sat}}^{\text{sat}} \propto \ln(N)$ uncover a link between two qualitatively different resources theories: the coherence in computational basis [Phys. Rev. Lett. 113, 140401 (2014)], related to the collision probability, and the magic resource theory.

Minor Comments and Questions:

1. I think the term "entropy" is somewhat misleading since the quantity is the logarithm of an expectation value, and it only becomes a probability distribution for $d = 2$.

We thank the Reviewer for this question. The name entropy is motivated by the similarity of the introduced measure with the SRE. Nevertheless, the quantity in the logarithm, i.e. the generalized stabilizer purity ζ_W , is indeed related to a probability distribution only in specific cases. For this reason, in the revised manuscript, instead of CSS entropy, we use the name of generalized stabilizer entropy (GSE) to highlight this analogy. We have expanded the discussion of this point in Sec. "Generalized stabilizer entropies" in the revised manuscript to avoid any confusion about this point.

2. Why are two different quantities used in the second figure and the insets of Figs. (2) and (3)? I think that using $\Delta Y/N$ can be quite misleading when making a direct comparison of the curves.

We thank the Referee for the question. The inset demonstrates the scaling collapse of the data. We preferred to include both ΔY and $\Delta Y/N$ (or $\Delta M_W(t)$ and $\Delta M_W(t)/N$ in the denotations of the revised manuscript) in the inset for completeness of the presentation and to highlight the scaling $\Delta M_W \sim N e^{-\alpha_d w t}$.

Figure 3: Convergence of the tensor network contraction for $\tilde{M}_2(t)$ for systems of $N = 32$, (a), and $N = 64$, (b) qubits with the bond dimension χ of the employed matrix product state. Results for the effective non-unitary dynamics for a system with the effective on-site Hilbert space dimension $q_{\text{eff}} = 4! = 24$, (a), and after reducing the on-site Hilbert space to $q'_{\text{eff}} = C_4 = 14$ dimensions, (b).

Reply to Reviewer 2

In the paper the authors study the behavior of magic in random quantum circuits. Using replica methods combined with tensor network numerics, they are able to systematically study the behavior of magic in local random circuits acting on qubits and qutrits up to hundreds or thousands of sites. The result that magic saturates in log-depth even in 1D circuits is physically reasonable and a clear intuition is provided for this behavior which sharply contrasts with entanglement entropy. The numerical results are also impressive as the solution of the replica theory is quite challenging for more than 3 replicas.

One comment on the analysis of qubits is that the effective dimension for 4 replicas is actually less than 24 because and gets reduced to 14. This point could be used to improve the numerical results. In addition, there are many symmetries in the replica model that should allow further reductions in the local dimension. I believe the results for qubits could be improved substantially by taking advantage of these effects.

Overall, I find the paper timely, accurate and clearly presented and recommend publication in Nature Communications.

We thank the Referee for their careful reading, the positive assessment of our work, and the insightful suggestions.

We followed the advice of the Referee and implemented for qubits ($d = 2$) the $k = 4$ replica tensor network utilizing the irreducible representations. This, as noted also in [Quantum Mach. Intell. 6, 54 (2024)], allows to reduce the effective Hilbert space dimension for k replicas from $q_{\text{eff}} = k!$ to $q'_{\text{eff}} = C_k$ where C_k is the k -th Catalan number. Specifically, as anticipated by the Referee, for $k = 4$ and qubits, this enables the reduction of the on-site Hilbert space dimension from $q_{\text{eff}} = 24$ to $q'_{\text{eff}} = C_4 = 14$. We calculated the $k = 4$ representation of the single qubit Haar random gate to find the tensors involved in the resulting tensor network contraction. We verified that this gate is a projector onto the 14-dimensional subspace of the 24-dimensional on-site Hilbert space left invariant by the two-qubit gates. This enabled us to construct the transformation which maps the tensors living in the $q_{\text{eff}} = 24$ -dimensional space to the invariant $q'_{\text{eff}} = 14$ subspace. We used this transformation to map the two-body gates, as well as

the initial and the final states involved in the tensor network contraction, to the $q'_{\text{eff}} = 14$ dimensional subspace.

The reduction $q_{\text{eff}} = 24 \rightarrow q'_{\text{eff}} = 14$ speeds up the calculations in two crucial ways. Firstly, the size of the tensors $\mathcal{T}_{i,i+1}^{(k)}$, corresponding to the two-body gates, is reduced from $q_{\text{eff}}^2 \times q_{\text{eff}}^2$ to $(q_{\text{eff}})'^2 \times (q'_{\text{eff}})^2$. The computational cost of contracting $\mathcal{T}_{i,i+1}^{(k)}$ with the tensors of the matrix product state is dominated by the SVD and scales as $(q'_{\text{eff}} \chi)^3$ (as detailed in Methods of our manuscript). Therefore, the reduction of the on-site Hilbert space dimension, for a *fixed value* of χ provides, for $k = 4$, a speed up by a factor of $(q'_{\text{eff}}/q_{\text{eff}})^3 \approx 5$. However, this is not the only factor speeding up the calculation. As shown in Fig. 3, the bond dimension χ required to reach the convergence of calculations is significantly smaller for the model with the reduced q'_{eff} -dimensional on-site Hilbert space. The data are well converged for $\chi_{\text{conv}} = 800$ in Fig. 3(a) and for $\chi_{\text{conv}} = 192$ in Fig. 3(b). Therefore, the overall speed up of the computation is $(q_{\text{eff}} \chi_{\text{conv}})^3 / (q'_{\text{eff}} \chi'_{\text{conv}})^3 \approx 360$, while the required memory is smaller by a factor of $(q_{\text{eff}} \chi_{\text{conv}})^2 / (q'_{\text{eff}} \chi'_{\text{conv}})^2 \approx 50$.

The reduction to the $q'_{\text{eff}} = 14$ model allowed us to substantially improve the numerical data for the stabilizer Rényi entropy of qubits and to reach system size $N = 1024$. The above considerations highlight the importance of utilizing the symmetries of the model to simplify computations. We comment on this crucial simplification for stabilizer Rényi entropy of qubits in the Methods of the revised manuscript.

Summary of the main changes in the revised manuscript

1. To clarify the focus of the manuscript, i.e., the phenomenology of magic spreading under local many-body dynamics, we:
 - reduce the technical description of the GSE (which is the new name for the CSS entropy), delegating the details from the Main Text and Methods to the Supplementary Material
 - highlight GSE as a tool for an efficient computation of magic resources growth under random quantum circuit dynamics
 - emphasize the parallels between GSE M_Y growth and the behavior of SRE for qutrits and introduce to the Main Text a new Fig. 3 which presents time evolution of SRE for qutrits.
2. We provide a detailed numerical investigation of the self-averaging of the SRE and GSE in the Methods of the raised manuscript.
3. We include a proof of the monotonicity of SRE for any qudit dimension $d > 2$ in the Supplementary Material. We construct a counter example proving that the GSE M_Y is not a monotone when the computational basis measurements are considered.
4. In the revised Methods, we discuss the improvement of the replica tensor network contraction for qubits which enables a significant reduction of the on-site Hilbert space dimension in the effective model, as suggested by the Reviewer 2.
5. We include numerical results for growth of mana under Haar random brick-wall circuits acting on qutrits in the revised Method. In the Supplementary Material, we provide numerical results for SRE for qubits with indices $q = 1, 3, 4$.

Reply to Reviewer 1

I have reviewed the revised materials and appreciate the authors' thorough efforts in addressing the referee's comments. Most suggestions have been effectively incorporated, with valuable additions that enhance the manuscript. Overall, its quality has significantly improved.

I carefully considered the revised materials in assessing this work's suitability for Nature Communications. From the outset, I found the topic and many of the results conceptually interesting. The authors have done an excellent job exploring and presenting their findings, especially after this revision.

However, my critical comments on the work's strength and impact remain, particularly in comparison to the stringent standards of Nature Communications. While I acknowledge the authors' effort in analyzing the spreading of magic in ergodic systems, I do not find the results sufficiently significant for this journal. That said, I believe the paper could be a valuable contribution to other, more specialized journals.

We thank the Referee for the positive assessment of our work.

Below are some additional comments.

- I suggest highlighting the technical result on the monotonicity of the stabilizer Rényi entropies, as the techniques may be of interest to the community.

We have followed the Referee's suggestion further highlighted our result on the monotonicity of the stabilizer Rényi entropy in the Methods section.

- There is a missing citation in the Conclusions near the sentence: "Steps in that direction were already taken for integrable systems."

We thank the Referee for pointing this out. We have solved this issue.

Reply to Reviewer 2

The authors have made several nice changes to the manuscript in response to the referee comments. I recommend the paper for publication in Nature Communication.

We thank the Referee for the positive assessment of our work.